# Repeated Tick Infestations Impair *Borrelia burgdorferi* Transmission in a Non-Human Primate Model of Tick Feeding

**DOI:** 10.3390/pathogens12010132

**Published:** 2023-01-13

**Authors:** Sukanya Narasimhan, Carmen J. Booth, Mario T. Philipp, Erol Fikrig, Monica E. Embers

**Affiliations:** 1Section of Infectious Diseases, Department of Internal Medicine, Yale University School of Medicine, New Haven, CT 06510, USA; 2Department of Comparative Medicine, Yale University School of Medicine, New Haven, CT 06520, USA; 3Division of Bacteriology & Parasitology, Tulane School of Medicine, New Orleans, LA 70112, USA

**Keywords:** non-human primate, *Borrelia burgdorferi*, Ixodes scapularis, acquired tick resistance

## Abstract

The blacklegged tick, *Ixodes scapularis*, is the predominant vector of *Borrelia burgdorferi*, the agent of Lyme disease in the USA. Natural hosts of *I. scapularis* such as *Peromyscus leucopus* are repeatedly infested by these ticks without acquiring tick resistance. However, upon repeated tick infestations, non-natural hosts such as guinea pigs, mount a robust immune response against critical tick salivary antigens and acquire tick resistance able to thwart tick feeding and *Borrelia burgdorferi* transmission. The salivary targets of acquired tick resistance could serve as vaccine targets to prevent tick feeding and the tick transmission of human pathogens. Currently, there is no animal model able to demonstrate both tick resistance and diverse clinical manifestations of Lyme disease. Non-human primates serve as robust models of human Lyme disease. By evaluating the responses to repeated tick infestation, this animal model could accelerate our ability to define the tick salivary targets of acquired resistance that may serve as vaccines to prevent the tick transmission of human pathogens. Towards this goal, we assessed the development of acquired tick resistance in non-human primates upon repeated tick infestations. We report that following repeated tick infestations, non-human primates do not develop the hallmarks of acquired tick resistance observed in guinea pigs. However, repeated tick infestations elicit immune responses able to impair the tick transmission of *B. burgdorferi*. A mechanistic understanding of the protective immune responses will provide insights into *B. burgdorferi*-tick–host interactions and additionally contribute to anti-tick vaccine discovery.

## 1. Introduction

The blacklegged tick, *Ixodes scapularis*, is the predominant vector of *Borrelia burgdorferi*, the agent of Lyme disease in the USA [1]. In addition, *I. scapularis* transmits other human pathogens, including *Babesia microti*, Powassan virus, *Anaplasma phagocytophilum*, and *Borrelia miyamotoi* [2,3]. Multipronged approaches to control tick populations and infection prevalence in endemic areas and to develop vaccines to prevent the tick transmission of these pathogens for human use are being actively pursued [4]. A vaccine targeting the outer surface protein A (OspA) of *B. burgdorferi* to prevent the transmission of *B. burgdorferi* to humans was developed almost two decades ago and marketed by Smith Kline Beecham [5,6]. However, the vaccine was removed from the market, in part due to a contentious hypothesis that the OspA vaccine might elicit T-cell cross-reactivity to an OspA epitope and provoke arthritis in some vaccine recipients [6,7,8]. That the OspA vaccination resulted in detrimental side effects remains to be validated [8,9,10,11,12]. A next-generation OspA-based vaccine that includes the immunoprotective epitopes of OspA from multiple strains of *B. burgdorferi* [13] is currently being developed by Pfizer Inc. in collaboration with Valneva SE Specific pathogen-targeted vaccines such as the OspA- or OspC-based [14] vaccines prevent the transmission of *B. burgdorferi* but cannot prevent the transmission of other tick-transmitted pathogens. Ticks can be coinfected with multiple pathogens; hence, a tick bite can transmit more than one pathogen simultaneously [2]. Feeding is fundamental to pathogen transmission, and tick salivary proteins are critical for tick feeding [15]. Tick salivary proteins may, therefore, serve as vaccine targets to prevent tick feeding [16,17]. 

Technological advances in DNA, RNA, and protein sequencing have accelerated our understanding of the tick salivary transcriptome and proteome over the last few years [18,19,20]. However, the subset of critical salivary antigens that may be ideal vaccine antigen targets remains elusive. One approach to determine the critical subset of salivary antigens is to exploit the phenomenon of acquired tick resistance or ATR, first described by Trager in 1939 [20]. Trager observed that natural hosts such as mice can be repeatedly infested by *I. scapularis* without acquiring tick resistance [21]. However, upon repeated tick infestations, non-natural hosts such as guinea pigs, mount a robust immune response against critical tick salivary antigens, and acquired tick resistance thwarts tick feeding. Nazario et al. [22] showed that the repeated infestations of guinea pigs with *I. scapularis* nymphs elicited ATR and also prevented the tick transmission of *B. burgdorferi*. The salivary targets of acquired tick resistance may, therefore, represent the critical subset of salivary antigens and could potentially serve as vaccine targets to prevent tick feeding and consequently prevent the tick transmission of human pathogens. Indeed, the immunization of guinea pigs with adult tick saliva, representing the salivary antigens secreted into the tick-bite site, recapitulated the hallmarks of tick resistance, including erythema at the tick bite site, the rapid rejection of ticks, and impaired tick feeding [23]. However, all parameters of Lyme disease could not be assessed since the guinea pig is not a robust model of Lyme disease [24], with its *B. burgdorferi* infection limited predominantly to the skin. Burke et al. [25] suggested humans may develop resistance to ticks upon repeated infestations, and this may have a consequence on Lyme disease incidences. Currently, an animal model able to demonstrate both acquired tick resistance and Lyme disease does not exist. Non-human primates (NHPs) serve as robust models of human Lyme disease [26,27]. However, it is not known whether non-human primates develop tick resistance upon repeated tick infestations. In this study, we assessed the development of acquired tick resistance in non-human primates upon repeated tick infestations and the impact on *B. burgdorferi* transmission. 

## 2. Materials and Methods

### 2.1. Ethics Statement for Animal Use 

Practices in the housing and care of nonhuman primates conformed to the regulations and standards of the Public Health Service Policy on Humane Care and Use of Laboratory Animals, and the Guide for the Care and Use of Laboratory Animals. The Tulane National Primate Research Center (TNPRC) is fully accredited by the Association for the Assessment and Accreditation of Laboratory Animal Care International. The Tulane University Institutional Animal Care and Use Committee approved all animal-related protocols, including the tick infestation and sample collection from NHPs. All animal procedures were performed or overseen by the TNPRC Division of Veterinary Medicine veterinarians and their staff. NHPs were always pair-housed, except when tick containment devices and jackets were in use; for the experimental period, paired monkeys were in protected contact. Rhesus macaques received food (monkey chow) and water ad libitum, and standard enrichment (food supplements, manipulatable items in the cage, human interaction with caretakers, perches, or swings). Routine husbandry practices included the reporting of any abnormal clinical signs or activity by NHPs to the appropriate veterinary medical staff and researchers responsible for the NHPs on this project.

### 2.2. Generation of Ixodes Scapularis Pathogen-Free and B. burgdorferi-Infected Nymphs

*I. scapularis* larvae were obtained from the Connecticut Agriculture Experiment Station and used to generate *B. burgdorferi*-infected nymphs. To generate the *B. burgdorferi*-infected nymphs, two specific pathogen-free (SPF) female C3H/HeNCrl (C3H/HeN) mice (4–5 weeks old) (Charles River, MA) were needle-inoculated with *B. burgdorferi* (N40) intradermally, and infection was confirmed with the quantitative (q)PCR of skin punch biopsy at day 21 post-infection, as described earlier [28]. The infected mice were each fed upon by ~300 larvae. Repleted larvae were placed in mesh-top containers, placed in the incubators maintained at 23 °C and 85% relative humidity under a 14 h light–10 h dark photoperiod, and larvae were allowed to molt to nymphs. The total DNA was isolated from 15–20 individual nymphs using the DNeasy Blood And Tissue Kit (Qiagen, Hilden, Germany), and the *B. burgdorferi* burden was assessed with qPCR, as described earlier [28], to estimate infection frequency. Routinely, we observed a 90–95% infection rate, and these were used for transmission experiments. 

To generate pathogen-free nymphs, larvae were fed to repletion on SPF C3H/HeN mice, and the repleted larvae were allowed to molt to nymphs as described above. All manipulations were conducted by strictly adhering to BL2 procedures. 

### 2.3. Tick Infestation of Rhesus Monkeys

Two 3-year-old male Rhesus monkeys (Macaca mulatta) of Chinese origin were acquired from the specific pathogen-free colony at TNPRC. The animals were anesthetized and challenged with 20 uninfected *I. scapularis* nymphs (10 on the right shoulder and 10 on the left shoulder), and ticks were allowed to feed to repletion using tick containment capsules protected by tampering-proof vests, as described earlier [27]. For blood collection, biopsy collection, and the tick capsule procedures, the monkeys were anesthetized with ketamine (10 mg/kg) or telazol (0.06–0.1 mL/kg) via intramuscular injection. Blood was collected through the femoral vein. Skin biopsies were taken using a sterile 4 mm punch, and the wound was sutured by the veterinarian. Veterinarians also provided analgesic drugs, such as buprenorphine after biopsies. The unfed nymphs were weighed in pools of 10 ticks prior to placement on the NHPs. After repletion, the NHPs were anesthetized, and the capsules were removed to retrieve the ticks. The ticks were weighed in pools of 3–5, and the average weight of the ticks was calculated. Three rounds of tick infestations were conducted with a 2-week resting period between each infestation. The ticks were weighed prior to placement on the animals and after repletion as described above. The tick bite sites were examined for any signs of erythema. 

### 2.4. Tick Transmission of B. burgdorferi 

To assess *B. burgdorferi* transmission, 10 *B. burgdorferi*-infected nymphs were placed on each of the 2 NHPs that had been infested by *I. scapularis* pathogen-free nymphs 3 times. Two age-matched control NHPs were similarly challenged with the 10 *B. burgdorferi*-infected nymphs. Tick placement was performed as described above for the placement of pathogen-free ticks, and the ticks were allowed to feed to engorgement. The engorged ticks were weighed in pools of 3–5, and pools of 2–3 ticks were processed for genomic DNA purification using the DNeasy Blood and Tissue Kit (Qiagen). *B. burgdorferi flaB* amplification was assessed with qPCR as described earlier [29], and the data were normalized to actin.

### 2.5. Skin Biopsy, Histology, Necropsy, and B. burgdorferi Evaluation

At 2 and 6 weeks post-tick challenge with the *B. burgdorferi*-infected nymphs, 4 mm skin punch biopsies from the NHPs were again obtained as described. For the skin biopsies, the NHPs were anesthetized, skin biopsies were collected using a sterile 4 mm biopsy punch, and the skin biopsy sites were sutured by a member of the clinical veterinary staff. 

After 24–48 h, the 2-week biopsies were washed in tap water for 30 min and then placed in an automatic processor, embedded in paraffin, and 5 um sections were generated for routine hematoxylin-and-eosin staining or Toluidine blue staining as described earlier [30]. The slides were analyzed by investigators blind to the experimental treatment (CJB, Yale University Comparative Pathology Research Core) using a Zeiss AxioScope microscope and imaged using an AxioCam MRC Camera and AxioVision 4.5 imaging software (Carl Zeiss Microimaging, Inc., Thornwood, NY, USA) and optimized in Adobe Photoshop (version 10.0, Adobe Systems Inc., San Jose, CA, USA). To assess the severity of the inflammation, a semiquantitative scoring method was used as described earlier [31].

At week 6 post-tick infestation, the animals were humanely euthanized with ketamine hydrochloride (10 mg/kg), followed by an overdose with sodium pentobarbital and submitted for necropsy (TNPRC). This method is consistent with the recommendation of the American Veterinary Medical Association guidelines. The heart base and pericardium, ½ half of the skin biopsies, and the joint tissues including the capsule and synovium were snap-frozen and subsequently processed for the total genomic DNA using the DNAeasy Genomic DNA Isolation Kit (Qiagen). *B. burgdorferi flaB* amplicons were assessed with qPCR as described earlier [29], and the data were normalized to actin. 

The second half of the skin biopsies taken at necropsy were dropped into a 5 mL BSK-H medium (Sigma, Saint Louis, USA) and cultured over 10 days at 33 °C for 10 days for the growth of *B. burgdorferi.* The spirochetes were visualized via dark-field microscopy, and the presence of viable motile spirochetes was scored as a positive culture. 

### 2.6. Skin Biopsy, Necropsy, and B. burgdorferi Evaluation

At week 2 post-tick challenge with *B. burgdorferi*-infected nymphs, the NHPs were anesthetized, and 4 mm skin punch biopsies were again obtained aseptically. Half of the skin biopsy was dropped into a 5 mL BSK-H medium (Sigma) and cultured over 10 days at 33 °C for 10 days for the growth of *B. burgdorferi*. The spirochetes were visualized via dark-field microscopy, and the presence of viable motile spirochetes was scored as a positive culture. At week 6 post-tick infestation, the animals were humanely euthanized and necropsied. The heart, skin, and joints were snap-frozen and subsequently processed for the total genomic DNA using the DNAeasy Genomic DNA Isolation Kit (Qiagen). *B. burgdorferi* flaB amplicons were assessed using qPCR as described earlier [29], and the data were normalized to actin. 

### 2.7. Statistical Analysis

The significance of differences between the groups was assessed with ordinary one-way ANOVA and Tukey’s post hoc test. 

## 3. Results

### 3.1. Repeated Infestation of Non-Human Primates with I. scapularis Does Not Elicit the Hallmarks of Acquired Tick Resistance

Two male Rhesus monkeys (Macaca mulatta) were challenged with 20 pathogen-free *I. scapularis* nymphs, and ticks were allowed to feed to repletion, as described in Materials Methods and shown in Figure 1, using tick containment capsules protected by tamper-proof vests as described earlier [27]. Three rounds of tick infestations were conducted, with a two-week resting period between each infestation. Although fewer ticks were repleted at the second infestation, the percentage of replete ticks was not significantly different at subsequent infestations (Figure 2A). The engorgement weights of the ticks were comparable across all three infestations (Figure 2B). The fold increase in engorgement weights (the average engorgement weights compared with the average unfed tick weights) at each infestation was comparable across all three infestations (Figure 2C). No visible erythema was observed at the tick bite sites at the first, second, or third infestations.

### 3.2. Tick Transmission of B. burgdorferi Is Impaired on Three-Time Tick-Infested Non-Human Primates

Two weeks after the third infestation, the NHPs were challenged with 10 *B. burgdorferi* (N40)-infected nymphal ticks generated as described in Materials and Methods and shown in Figure 1. Two age-matched naïve NHPs were similarly challenged with *B. burgdorferi*-infected nymphs, and ticks were allowed to feed to repletion. The engorgement weights, fold increase in engorgement, and the percentage of repleted ticks were comparable between the naïve and three-time tick-infested NHPs (Figure 3). 

The *B. burgdorferi* burden in repleted ticks was evaluated using quantitative PCR, as described in Materials and Methods, and the results showed a comparable burden in the ticks fed on naïve or tick-infested NHPs (Figure 4A). At two weeks post-tick challenge, skin punch biopsies were obtained from naive and tick-infested NHPs and assessed for *B. burgdorferi* burden using culture and quantitative PCR. The *B. burgdorferi* burden in the skin of tick-infested NHPs was decreased compared with the skin of naïve NHPs (Figure 4B). The cultures were negative for *B. burgdorferi* in both three-time tick-infested NHPs and positive in both naïve NHPs (Table 1). The NHPs were humanely euthanized 6 weeks post-tick challenge, and *B. burgdorferi* dissemination into the heart, bladder, joints, and skin was assessed with culture and quantitative PCR, as described in Materials and Methods. *B. burgdorferi* burdens were significantly decreased in the skin and heart tissues of the three-time tick-infested NHPs compared with that in naïve NHPs (Figure 4B). While we could not culture *B. burgdorferi* from the skin of the three-time tick-infested NHPs, all other tissues of these three-time tick-infested animals were culture-positive (Table 1). All tissues of naïve NHPs were culture-positive (Table 1).

### 3.3. Histological Assessment of Skin Biopsies Demonstrates Increased Inflammation at the Tick Bite Site upon Repeated Tick Infestations

The skin biopsies taken aseptically from the tick bite sites from naïve or three-time tick-infested NHPs were immersion-fixed in a 10% solution of zinc-modified formalin and processed to hematoxylin-and eosin-stained slides using routine methods. The slides were examined by investigators blind to the experimental treatment and scored by performing a semiquantitative analysis, as outlined in the Materials and Methods. The sections of the skin from naïve NHPs were unremarkable, with scant to minimal dermal and perivascular lymphocytes and plasma cells, consistent with normal dermis (Figure 5A; left lower panel, arrowheads). In contrast, the skin from the three-time tick-infested NHPs had an overall mild to moderate, and in one section, a marked increase in perivascular, superficial, and deeper dermal lymphocytes and plasma cells (Figure 5A; right panels). Further, there was epidermal hyperplasia and dermal edema within the areas of tick-bite inflammation. We also observed neutrophils and eosinophils within the dermis and perivascular sites and the margination of neutrophils along vessel walls in some sections (not shown). The semiquantitative scoring as described in Materials and Methods showed an increased inflammatory milieu in the three-time tick-infested NHPs compared with naïve NHPs (Figure 5B). In our review of the T-Blue-stained sections for mast cells and basophils, they were found unremarkable. 

## 4. Discussion

Trager’s seminal observation showed that the non-natural hosts of Ixodes ticks develop resistance to ticks upon repeated infestations, resulting in erythema at the tick bite sites and impaired tick feeding, as evidenced by the rapid rejection of ticks and decreased engorgement weights [21]. This has since been recapitulated by several studies using different tick–host models [22,32,33,34,35,36,37,38]. The recruitment of cellular and humoral immune responses to secreted salivary antigens critical for tick feeding can be invoked in the elicitation of tick resistance [38,39,40,41,42]. This phenomenon has since been exploited by research efforts in pursuit of salivary antigens that may serve as anti-tick vaccine targets [16]. Our focus is centered on *I. scapularis*, which serves as a vector of multiple human pathogens [2]. Attesting to the critical role of immunity against salivary antigens in acquired tick resistance, the immunization of guinea pigs with tick saliva collected from engorged adult ticks provided erythema at the tick bite sites and impaired tick feeding [23]. However, unlike ATR, tick rejection was observed only at about 48 h post-tick attachment [23]. The salivary transcriptome of *I. scapularis* dynamically changes in composition throughout the course of engorgement [30,43]. Earlier studies showed that the salivary proteins represented in the first 24 h of tick attachment are sufficient to elicit the hallmarks of tick resistance, to reject ticks within 24 h of tick attachment, and to impair the tick transmission of *B. burgdorferi* [30]. Similarly, recent work by Lynn et al. [44] showed that the antigens in tick cement, representing the salivary antigens expressed early in feeding, may also serve as anti-tick vaccine targets. However, the critical subset of salivary antigens that can serve as a vaccine cocktail to thwart tick feeding remains elusive. Given the functional redundancy in the tick salivary proteome [19,45], there is a consensus that a single salivary antigen is unlikely to serve as a potent vaccine. Therefore, a cocktail of multiple critical antigens is warranted. Difficulties in generating recombinant tick salivary proteins with post-translational modifications, the choice of the optimal subset of antigens, and the choice of adjuvant present some of the challenges plaguing anti-tick vaccine development. Exploiting a novel mRNA–lipid nanoparticle (LNP) delivery platform, Sajjid et al. [46] showed that the immunization of guinea pigs with an mRNA–LNP cocktail of 19 *I. scapularis* salivary antigens (19IsP), a subset of which were also targeted by tick-resistant rabbit or guinea pig sera, provided erythema at the tick bite site and robust tick rejection within 48 h of tick attachment. Initial studies to determine if immunity against 19IsP would also impair the tick transmission of *B. burgdorferi* suggested that the removal of ticks at the onset of erythema at the tick bite site prevented transmission. Interestingly, the 19IsP vaccination of Mus musculus, a laboratory model of a natural host, failed to elicit tick rejection [46]. Mus musculus do not develop ATR upon repeated tick infestations [47,48]. While the guinea pig is a robust model of ATR, only the initial phase of *B. burgdorferi* transmission can be evaluated using the guinea pig model of Lyme disease since *B. burgdorferi* infection is limited to the skin [24]. An animal model able to recapitulate ATR and human Lyme disease would accelerate anti-tick vaccine development. The NHP model of Lyme disease is fully reflective of human Lyme disease [27], but its utility as a model of ATR remains unknown. 

Repeated tick infestations had no significant impact on tick feeding success on the NHP model with nymphal tick engorgement comparable in naïve or repeatedly tick-infested NHPs. However, the skin biopsies taken from the tick bite site after the fourth tick infestation revealed significantly increased inflammation compared with that in naïve NHPs at the first tick infestation. The inflammatory milieu was predominantly neutrophilic and eosinophilic. Studies on guinea pig skin biopsies obtained at the third or fourth tick infestation have shown the predominant recruitment of basophils, mast cells, and eosinophils [30,49,50,51,52]. It has been suggested that the degranulation of basophils results in the release of noxious components, including proteases and histamine that deter tick feeding and promote early tick rejection and thus impairment of the tick transmission of pathogens [53]. The repeated tick infestations of NHPs did not result in tick rejection, potentially due to the lack of a basophilic milieu at the tick bite sites. The increased presence of neutrophils at the three-time tick-infested NHP skin was reminiscent of three-time tick-infested mouse skin biopsies [47,48] and contrasted with the basophilic milieu observed at the tick bite sites on tick-resistant guinea pig skin. Nevertheless, the repeated tick infestations of NHPs resulted in the impaired tick transmission of *B. burgdorferi* despite comparable *B. burgdorferi* burden in engorged ticks. Neutrophils have been shown to be detrimental to *B. burgdorferi* by virtue of their ability to release reactive oxygen species through oxidative burst [54,55] and through the formation of neutrophil extracellular traps that can retain and trap the spirochetes within this net [56]. Tick saliva is known to inhibit neutrophil functions [57], and this could serve to defuse the detrimental impact of neutrophils on spirochete survival. This suggests NHPs develop an immune response to the tick salivary proteins that may function to defuse neutrophil recruitment and activation and potentially account for impaired *B. burgdorferi* transmission and survival in the mammalian host. This is also consistent with earlier studies that showed that repeated infestations of the murine host result in the rapid and increased recruitment of neutrophils to the tick bite site [47] and impairment of the tick transmission of *B. burgdorferi* [58]. While the tick-infested NHP sera did not provide conclusive immunoreactivity to tick salivary protein extracts, we cannot rule out the possibility that the NHPs may have developed neutralizing antibodies to salivary proteins such as Salp15, shown to enhance *B. burgdorferi* transmission to the mammalian host [59].

## 5. Conclusions

We recognize that the tick sialome is complex, encoding functional paralogs [19], changing in composition temporally during tick feeding [30,43], and potentially also changing in composition on different host species [47]. The varying extent of ATR in different host species [53] also confounds the utility of ATR to define the critical salivary proteins that may serve as anti-tick vaccine targets for human use. Advances in molecular techniques to decipher the tick sialome and to dissect the mechanisms of ATR will help circumvent these limitations. Our results suggest that NHPs may not generate significant ATR to *I. scapularis* nymphal ticks. Whether the repeated infestations of NHPs with adult *I. scapularis* ticks may generate ATR remains to be determined. Nevertheless, our observations suggest that the NHP model of tick-borne Lyme disease may not only help define the tick salivary antigens that are critical for the tick transmission of *B. burgdorferi* but may also serve as a model for other tick-borne pathogens [60,61,62] and open a new avenue to define the tick salivary antigens involved in the enhanced transmission of these tick-borne pathogens. Furthermore, the rhesus macaque model may most accurately predict the efficacy of anti-tick vaccines designed for humans. 

## Figures and Tables

**Figure 1 pathogens-12-00132-f001:**
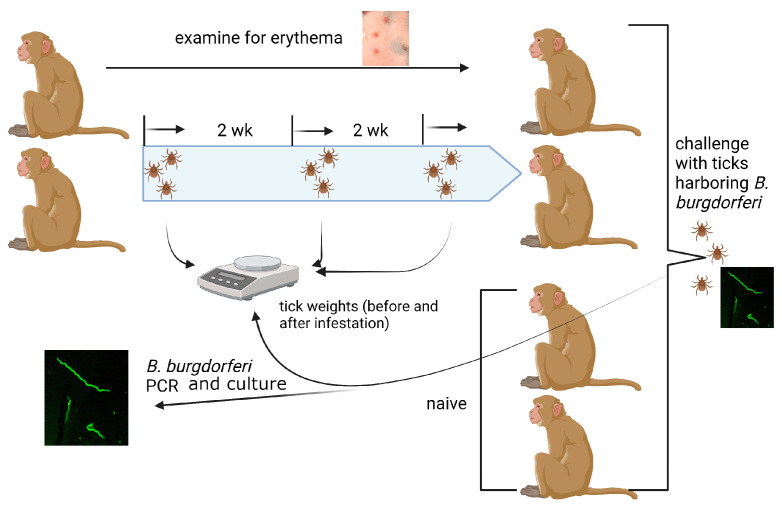
Experimental design to test ATR and resistance to pathogen transmission by repeated tick infestation of non-human primates (NHPs). Two rhesus macaques were infested with 20 pathogen-free *Ixodes scapularis* nymphs and ticks fed to repletion. Tick weights were recorded as a measure of feeding efficiency. Tick infestations were repeated 2 more times once every 2 weeks. After the 3rd infestation, two naïve and two 3x-tick infested NHPs were challenged with 10 *Borrelia burgdorferi*-infected nymphs, and infection was monitored with culture and PCR of skin biopsies at 2 weeks and of skin biopsies, heart, and joints at necropsy 6 weeks post-tick challenge.

**Figure 2 pathogens-12-00132-f002:**
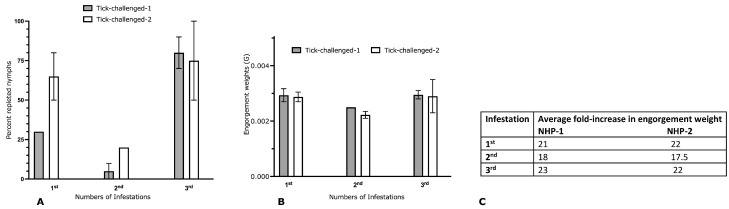
Repeated tick infestation of non-human primates does not impact tick engorgement. Two male Rhesus monkeys (Macaca mulatta) were subjected to three rounds of tick infestations with a 2-week resting period between each infestation and tick attachment, and engorgement was monitored at each infestation: (**A**) percentage of repleted ticks at each infestation; (**B**) average engorgement weight of ticks at each infestation; (**C**) fold increase in engorgement weights (average engorgement weights compared with average unfed tick weights) at each infestation.

**Figure 3 pathogens-12-00132-f003:**
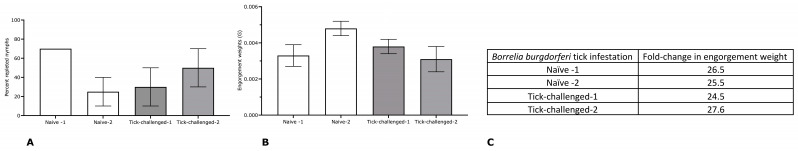
Engorgement success of *Borrelia burgdorferi*-infected nymphal ticks fed on naïve or 3x-tick infested non-human primates. Two male Rhesus monkeys (*Macaca mulatta*) subjected to three rounds of tick infestations or naïve were challenged with *B. burgdorferi*-infected nymphs, and engorgement success was monitored: (**A**) percentage of repleted ticks; (**B**) average engorgement weight of ticks; (**C**) fold increase in engorgement weights (average engorgement weights compared to average unfed tick weights).

**Figure 4 pathogens-12-00132-f004:**
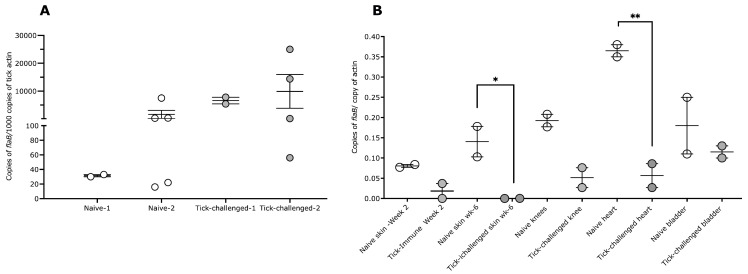
*Borrelia burgdorferi*-burden in repleted ticks and in naïve or 3x-tick infested non-human primates. *B. burgdorferi* burden evaluated with quantitative PCR: (**A**) repleted nymphs; (**B**) skin punch biopsies at 2 weeks post-tick challenge and in heart, bladder, joints, and skin at necropsy 6 weeks post-tick challenge of naive and 3x-tick-infested NHPs. Error bars are +SEM. Statistical significance was assessed using one-way ANOVA (*p* < 0.05 *; *p*< 0.005 **).

**Figure 5 pathogens-12-00132-f005:**
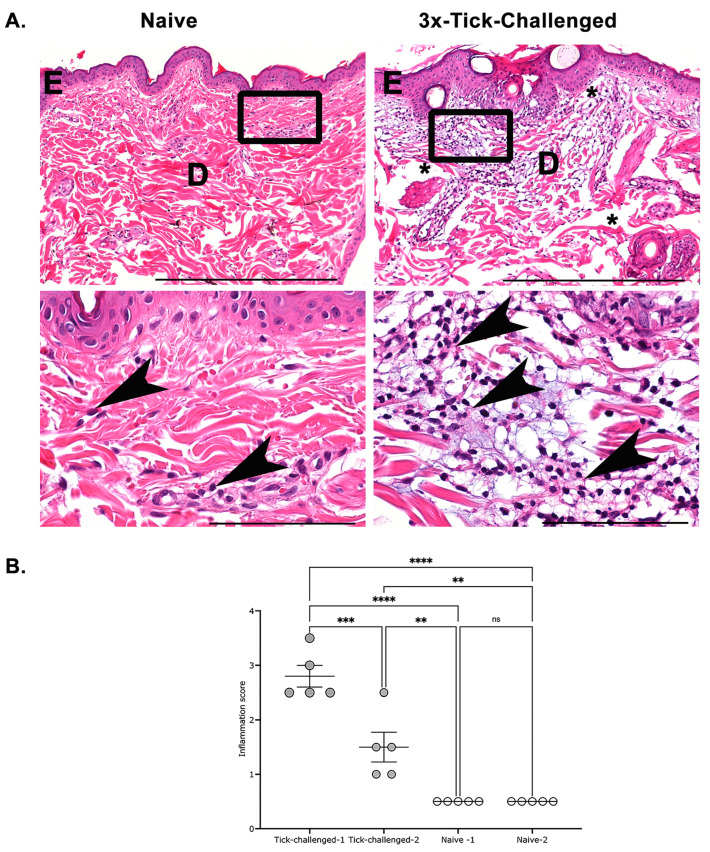
Representative photomicrographs from HE-stained section of 3x-Tick-challenged NHPs skin show increased inflammation over naive NHP skin: (**A**) naïve NHP Skin (left panels): representative photomicrographs of naive skin with unremarkable epidermis (E), dermis (D) and scattered perivascular lymphocytes and plasma cells (arrowheads); 3x-tick-challenged NHP Skin (right panels): representative photomicrographs show that the epidermis is slightly thickened (E), the dermis expanded by edema (*), and with a mild-to-moderate increase in dermal inflammatory cells (predominantly neutrophils and eosinophils (arrowheads) compared with naïve skin. (arrows). Upper panels scale bars = 500 µm, and lower panels scale bars = 100 µm; (**B**) semiquantitative scoring shows significantly increased inflammation in 3x tick-infested compared with naïve NHP skin. Error bars are mean + SEM. Mean values were significantly different using ANOVA with Tukey’s multiple-comparison test (*p* > 0.05 ns; *p* < 0.05 *; *p* < 0.005 **; *p* < 0.001 ***; *p* < 0.0001 ****).

**Table 1 pathogens-12-00132-t001:** Culture assessment of non-human primate tissues after *Borrelia burgdorferi*-infected tick challenge. Skin biopsies at 2 weeks and of skin, heart, joints, and bladder at 6 weeks post-*B. burgdorferi*-infected tick challenge of naïve or 3x-tick challenged non-human primates were assessed for viable *B. burgdorferi* spirochete growth in BSK-H complete medium.

Tissue	Naïve 1	Naïve 2	3x-Tick Challenged 1	3x-Tick Challenged 2
Skin-2 weeks post tick detachment	Positive	Positive	Negative	Negative
Skin-6 weeks post tick detachment	Positive	Positive	Negative	Negative
Heart	Positive	Positive	Positive	Positive
Joint/Knee	Positive	Positive	Positive	Positive
Bladder	Positive	Positive	Positive	Positive

## Data Availability

All available data from the project are presented within this manuscript.

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
