# Peer review of "Repeated Tick Infestations Impair Borrelia burgdorferi Transmission in a Non-Human Primate Model of Tick Feeding"

_pathogens, 2023, doi:10.3390/pathogens12010132_

Round 1

Reviewer 1 Report

The MS deals with an interesting and important topics, the effect of acquired tick resistance (ATR) on the transmission of tick-borne pathogens. The authors show that using non-human primates (Rhesus monkeys) as a model for studying ATR effect on Borrelia burgdorferi transmission by Ixodes scapularis ticks can be very useful. It may be close to the human situation, because humans can develop some ATR which can influence B. burgdorferi transmission.

The MS is well written, the results are convincing and well discussed. I have only one recommendation for Discussion. Possible mechanisms of ATR in Rhesus monkeys are not discussed. Is reduction of Borrelia spirochetes transmission caused by induced inflammatory reaction in the skin, or “neutralization” of immunomodulatory molecules (SAT factors) pays a role, or both mechanisms are involved? Even though the MS does not provide enough relevant data, mechanisms of ATR impact on tick-borne pathogen transmission should be discussed.

Author Response

We thank the reviewer for their careful review and for their helpful suggestions to improve the manuscript. We have incorporated the requested changes and our point-by-point response to the reviewers is below

Response: We have now discussed the possible explanation for the decreased B. burgdorferi transmission in NHPs repeatedly infested by ticks (Lines 336-338; 344-356)

Reviewer 2 Report

The well-designed experiments demonstrate the complexity and limited potential utility of tick-salivary proteins as vaccine candidates for the prevention of Lyme disease and other tick-associated diseases. The potential limited utility of salivary proteins as practical and effective vaccines to prevent tick-associated infections needs further emphasis in the discussion and conclusion sections.

In lines 38-40, the statement regarding the removal from the market of an OspA vaccine is not correct, and needs to be modified or deleted. That vaccine was effective, and it was not because of poor sales, but because there was legitimate concern that in some patients, more likely in those with a known or unknown history of Lyme disease, there were reactions resembling reactivation of prior Lyme disease-associated symptoms. The referenced articles do not disprove the legitimacy of these reactions and concerns, and concern remains about the potential adverse reactivity if given any form of an OspA vaccine. While a better understanding of the reactions that occurred has not been determined, as is not uncommon with some vaccines, eg mRNA COVID, it would be best to be more cautious with the statement used.

On line 120, delete the word "be"

Author Response

We thank the reviewer for their careful review and for their helpful suggestions to improve the manuscript. We have incorporated the requested changes and our point-by-point response to the reviewers is below.

Rev 2: The well-designed experiments demonstrate the complexity and limited potential utility of tick-salivary proteins as vaccine candidates for the prevention of Lyme disease and other tick-associated diseases. The potential limited utility of salivary proteins as practical and effective vaccines to prevent tick-associated infections needs further emphasis in the discussion and conclusion sections.

Response: We have addressed this in the Conclusion section (Lines 359-365).

In lines 38-40, the statement regarding the removal from the market of an OspA vaccine is not correct, and needs to be modified or deleted. That vaccine was effective, and it was not because of poor sales, but because there was legitimate concern that in some patients, more likely in those with a known or unknown history of Lyme disease, there were reactions resembling reactivation of prior Lyme disease-associated symptoms. The referenced articles do not disprove the legitimacy of these reactions and concerns, and concern remains about the potential adverse reactivity if given any form of an OspA vaccine. While a better understanding of the reactions that occurred has not been determined, as is not uncommon with some vaccines, eg mRNA COVID, it would be best to be more cautious with the statement used.

Response: We agree and have now rephrased the statements (Lines 37-40)

On line 120, delete the word "be”

Response: We have deleted the word.

Round 2

Reviewer 1 Report

I agree with the revised MS.

Reviewer 2 Report

Thanks to the authors for the revisions.